# Differences in the Outcome of Patients with COPD according to Body Mass Index

**DOI:** 10.3390/jcm9030710

**Published:** 2020-03-05

**Authors:** Zichen Ji, Javier de Miguel-Díez, Christian Reynaldo Castro-Riera, José María Bellón-Cano, Virginia Gallo-González, Walther Iván Girón-Matute, Rodrigo Jiménez-García, Ana López-de Andrés, Virginia Moya-Álvarez, Luis Puente-Maestu, Julio Hernández-Vázquez

**Affiliations:** 1Pulmonology Service, Gregorio Marañón University General Hospital, 28007 Madrid, Spain; jdemigueldiez@telefonica.net (J.d.M.-D.); christianrcr4758@gmail.com (C.R.C.-R.); vgallogonzalez@gmail.com (V.G.-G.); walter_giron2@hotmail.com (W.I.G.-M.); luis.puente@salud.madrid.org (L.P.-M.); 2Research Support Service, Gregorio Marañón Health Research Institute, 28007 Madrid, Spain; bellon23@gmail.com; 3Public Health and Maternal and Child Health Department, Faculty of Medicine, Complutense University of Madrid, 28040 Madrid, Spain; rodrijim@ucm.es; 4Preventive Medicine and Public Health Teaching and Research Unit, Health Sciences Faculty, Rey Juan Carlos University, 28922 Alcorcón, Madrid, Spain; ana.lopez@urjc.es; 5Pulmonology Service, Lozano Blesa University Clinical Hospital, 50009 Zaragoza, Spain; vmoyalvarez@gmail.com; 6Pulmonology Section, Infanta Leonor University Hospital, 28031 Madrid, Spain; juliohernandezvazquez@hotmail.com

**Keywords:** chronic obstructive pulmonary disease (COPD), body mass index, survival, risk factor

## Abstract

Background: In chronic obstructive pulmonary disease (COPD), the “obesity paradox” is a phenomenon without a clear cause. The objective is to analyze the complications of COPD patients according to their body mass index (BMI). Methods: An observational study with a six-year prospective follow-up of 273 COPD patients who attended a spirometry test in 2011. Survival and acute events were analyzed according to the BMI quartiles. Results: A total of 273 patients were included. BMI quartiles were ≤24.23; 24.24–27.69; 27.70–31.25; ≥31.26. During the follow-up, 93 patients died. No differences were found in exacerbations, pneumonia, emergency visits, hospital admissions or income in a critical unit. Survival was lower in the quartile 1 of BMI with respect to each of the 2–4 quartiles (*p*-value 0.019, 0.013, and 0.004, respectively). Advanced age (hazard ratio, HR 1.06; 95% confidence interval, CI 1.03–1.09), low pulmonary function (HR 0.93; 95% CI 0.86–0.99), exacerbator with chronic bronchitis phenotype (HR 1.76; 95% CI 1.01–3.06), high Charlson (HR 1.32, 95% CI 1.18–1.49), and the quartile 1 of BMI (HR 1.99, 95% CI 1.08–3.69) were identified as risk factors independently associated with mortality. Conclusions: In COPD, low BMI conditions a lower survival, although not for having more acute events.

## 1. Introduction

Obesity is a highly prevalent disease, and one of the most common health problems in the world. It is associated with reduced life expectancy, and is an indicator of poor prognosis in many diseases [1,2,3]. In terms of the respiratory system, obesity is harmful to the lungs because adipose tissue is a source of proinflammatory factors [4].

Chronic obstructive pulmonary disease (COPD) usually progresses with comorbid conditions such as obesity or being underweight [5]. Surprisingly, obese patients with COPD survive longer than patients with COPD who are not obese, despite the cardiovascular risk and inflammatory burden associated with obesity. This phenomenon is known as the “obesity paradox” [6,7,8].

Several studies describe this paradox and attempt to clarify the mechanism underlying it [9,10], such as poorer prognosis owing to malnutrition and loss of muscle mass in thin patients with COPD [11], overestimation of the degree of obstruction in obese patients [12,13,14], and the presence of confounders [15]. In any case, the reason why obesity behaves as a protective factor in patients with COPD remains undetermined [16].

The objective of the present study is to describe differences in patient characteristics, consumption of health care resources, and survival according to body mass index (BMI).

## 2. Patients and Methods

### 2.1. Design

We performed a noninterventional observational study with a prospective follow-up in a hospital in Madrid, Spain. To be included in the study, patients had to be aged >40 years with a history of smoking (active or otherwise at inclusion and a pack-year index >10). They also had to be diagnosed with COPD according to the definition of the Global Initiative for Chronic Obstructive Lung Disease (GOLD) [17] and have been admitted to the hospital to undergo spirometry at the request of any specialist and for any reason during the first half of 2011. We excluded patients with a forced expiratory volume in the first second (FEV_1_) >70%, those with other nonobstructive diseases of the pulmonary parenchyma, and those participating in a clinical trial. Therefore, the study population comprised only patients with at least moderate obstructive disease. In order to avoid bias, we included all patients who fulfilled all of the inclusion criteria and none of the exclusion criteria during the abovementioned recruitment period.

### 2.2. Variables Collected

Variables were collected at two time points. The first was at recruitment, and the variables recorded were patient characteristics (including comorbidity indices, pharmacological treatment, and respiratory therapies). Data were collected again during follow-ups in order to determine the consumption of health care resources by patients and survival.

The variables for patient’s characteristics were age, sex, weight, height, BMI, date of inclusion, FEV_1_, and forced vital capacity (FVC) expressed both in milliliters and as a percentage of what was predicted. We also recorded smoking history, COPD phenotype, treatment of COPD, respiratory therapy, and comorbidity (Charlson and COPD-specific Comorbidity Test (COTE)).

The variables related to consumption of health care resources and patient outcome during follow-up were acquisition of pneumonia, number of episodes of pneumonia, care in the emergency department, number of visits to the emergency department, admission to the hospital, number of admissions to the hospital, exacerbations of COPD, number of exacerbations of COPD, admission to the intensive care unit (ICU), number of admissions to the ICU, date of last follow-up for survivors, and date of death for those who died.

Patients were followed until 1 April 2017. In the case of patients who died during the follow-up period, the date of death was considered to be the end of the follow-up. In the case of patients who survived, the end of the follow-up was considered the last time the patient was seen in the health care setting.

### 2.3. Classification According to COPD Phenotypes

Patients had to belong to one of the four COPD phenotypes, which were mutually exclusive, as follows: positive bronchodilator response, nonexacerbator, exacerbator with emphysema, and exacerbator with chronic bronchitis. The criteria for classifying patients as belonging to a specific phenotype were as follows: first, patients who achieved an increase in FEV_1_ > 200 mL (absolute) and 12% compared with FEV_1_ without bronchodilation were classified as positive bronchodilator response; next, all patients who had visited the emergency department less than twice for a respiratory complaint and did not require admission during the 12 months before inclusion were classified as nonexacerbators; lastly, the remaining patients were classified as exacerbators with emphysema or exacerbators with chronic bronchitis depending on the predominant symptom, radiological findings, and lung function data. Thus, patients with dyspnea as their predominant symptoms or with radiological evidence of emphysema or a low diffusing capacity for carbon monoxide (DLCO) were classified as exacerbators with emphysema. In contrast, patients with cough and expectoration as their predominant symptoms for at least three months per year during the two years immediately before inclusion were classified as exacerbators with chronic bronchitis.

In this study, we did not follow the criteria of Spanish guidelines on COPD [18], which define asthma-COPD overlap syndrome as an increase in FEV_1_ of more than 400 mL and 15% during the bronchodilator test, since we consider that this definition is highly specific but insufficiently sensitive.

### 2.4. Statistical Analysis

The presence or absence of a normal distribution was assessed using a histogram.

Normally distributed quantitative variables were expressed as mean (SD); non-normally distributed quantitative variables were expressed as median (IQR). Qualitative variables were expressed as frequencies.

Normally distributed qualitative variables were analyzed using analysis of variance, whereas non-normally distributed quantitative variables were analyzed using the Kruskal–Wallis test. Qualitative variables and proportions were compared using the chi-square test or Fisher exact test depending on the sample size.

Survival was analyzed using Kaplan–Meier plots. The log-rank test was used for the univariate analysis of the probability of death depending on the BMI quartile. In contrast, Cox regression analysis was used for multivariate analysis of survival adjusted for age, sex, FEV_1_ (absolute), phenotype, active smoking, Charlson comorbidity index, and BMI quartile. The proportionality of risk was verified for all of the variables included in the multivariate model.

Statistical significance was set at *p* < 0.05 (2-tailed) for all comparisons.

Stata, Version 15 was used to construct the survival plots, perform the multivariate analysis, and verify the proportionality of risks. The remaining statistical analyses were performed using SPSS, Version 20.

### 2.5. Ethics Committee and Informed Consent

The study was approved by the Ethics Committee of Hospital General Universitario Gregorio Marañón. In order to participate in the study, patients had to have signed an informed consent document at inclusion.

## 3. Results

The study population comprised 273 patients, who were followed for a median of 68.16 months. The mean age was 68 years, and 243 patients were men. The mean weight was 75.03 kg, and the mean height was 1.63 m; consequently, the mean BMI was 28.05 kg/m^2^. As for lung function, the mean FEV_1_ (absolute) was 1211.39 mL, the mean FEV_1_ (predicted) was 48.64%, the mean FVC (absolute) was 2362.67 mL, and the mean FVC (predicted) was 73.16%. At inclusion, 92 patients were active smokers. Patients were distributed by phenotype as follows: positive bronchodilator response, 71; nonexacerbators, 135; exacerbators with emphysema, 27; and exacerbators with chronic bronchitis, 40. The median Charlson comorbidity index was 2, and the median COTE was 1. As for treatment of COPD, 254 patients were receiving a long-acting muscarinic antagonist at inclusion, 242 were receiving a long-acting adrenergic β_2_-agonist, and 212 were taking inhaled corticosteroids. Respiratory therapy took the form of long-term home oxygen therapy in 91 patients, continuous positive airway pressure (CPAP) in 31, and bilevel positive airway pressure in 14 patients. During follow-up, 178 patients had at least 1 exacerbation of COPD, with a median of 1 episode; 77 patients had at least 1 episode of pneumonia, with a median of 0 episodes; 120 patients had visited the emergency department at least once but did not have to be admitted; and 150 were admitted. A total of 28 patients were admitted to the ICU at some point, with a median of 0 admissions for critical care; 93 patients died. Full descriptive data are shown in Table 1.

Patients were divided into four groups according to BMI quartile in order to ensure that sample size was homogeneous. Thus, quartile 1 had a BMI of ≤ 24.23, quartile 2 had a BMI of 24.24–27.69, quartile 3 had a BMI of 27.70–31.25, and quartile 4 had a BMI of ≥ 31.26. Table 2 shows the sample size and mean (SD) for each BMI group.

Table 3 shows comparisons for sex, phenotype, lung function, obstructive sleep apnea (OSA), active smoking, treatment of COPD, respiratory therapy, and comorbidity scales according to the BMI group. A greater proportion of patients with BMI in quartile 1 had the exacerbator with emphysema phenotype. Furthermore, no patients in this quartile had OSA and were therefore not receiving CPAP. Furthermore, a higher percentage of patients in quartile 1 were active smokers at inclusion. Lastly, patients in BMI quartile 1 had a lower FEV_1_ (absolute) value than patients in other quartiles.

Table 4 shows comparisons with respect to exacerbations, acquisition of pneumonia, and use of health care resources during follow-up according to BMI quartile. No statistically significant differences were observed for onset of events or number of events.

Figure 1 shows the survival curve stratified by BMI quartile. Survival was lower for patients in BMI quartile 1 than for those in other quartiles. This difference was statistically significant with respect to the other three quartiles. Table 5 shows the mean survival in each BMI quartile, the log-rank test result, and the *p*-value of quartile 1 with respect to the other quartiles.

Table 6 shows factors that were independently associated with mortality, namely, advanced age, low FEV_1_ (absolute), exacerbator with chronic bronchitis phenotype, high Charlson comorbidity index, and BMI quartile 1.

## 4. Discussion

The main conclusion of our study was that low BMI in patients with COPD acts as an indicator of poor outcome compared with patients with a higher BMI. This negative effect of low BMI was even observed in patients whose weight was normal. We recorded a statistically significant difference in survival but not in the risk of exacerbations, pneumonia, or in the need for medical care in hospital. Patients who were in the BMI quartile 1 survived for a shorter period, had an absolute FEV_1_ value lower than that of patients from the other quartiles, and were more often exacerbators with emphysema. In addition, there were more active smokers at inclusion. However, no statistically significant differences were observed with respect to pharmacological treatment. No patients in the first BMI quartile had OSA; therefore, no patients were being treated with CPAP. No differences were found in the Charlson comorbidity index or COTE, although a high Charlson index was found to be an independent risk factor for mortality.

While numerous studies focus on explaining the obesity paradox in COPD, no specific theory has been widely accepted to date [16].

Several authors point to lung function as a cause, arguing that patients with a high BMI have better lung function than those with a low BMI [12,19]. Our results are in part consistent with this theory, since patients in BMI quartile 1 had a lower FEV_1_ than patients in quartiles 2–4, although no difference with FVC was recorded. In contrast with this theory, other authors think that lung function is overestimated in obese patients [20].

It has also been proposed that this paradox is due to a physical limitation caused by greater intra-abdominal pressure on the thorax in obese patients, leading to lower total lung capacity [21]. Our data do not support this proposal, since we found no differences in FVC between the different BMI quartiles.

Nutritional status is more widely accepted [11,22]. It seems that survival of patients with COPD could be influenced by body composition, with a low fat-free mass indicating poor prognosis [19,23]. Given that our study was based on spirometry data, which only includes weight and height values, it is somewhat limited with respect to the analysis of nutritional status.

In contrast, the strengths of our findings include the fact that we analyzed potential confounders that receive little attention elsewhere, for example, comorbidities, COPD phenotypes, and consumption of health care resources. As for comorbidities, no significant differences were found between the BMI quartiles with respect to the Charlson comorbidity index or the COTE result. However, the multivariate analysis revealed that the Charlson index was an independent risk factor for mortality. As for COPD phenotypes, the exacerbator with chronic bronchitis phenotype was identified as an independent risk factor for mortality. Furthermore, we showed that poorer survival in patients with a low BMI is not due to a greater probability of exacerbations, acquisition of pneumonia, visits to the emergency department, hospital admissions, or ICU admissions.

This study has a limitation. Regarding the pharmacological treatment, no data on the combined therapy were collected; instead, data were collected on each therapy separately. This has only allowed us to make an analysis of each component of the treatment, but not the combination of them. The pharmacological treatment of COPD is constantly evolving; Matsunaga et al. [24] has pointed out the importance of a more comprehensive pharmacological treatment. To better understand the obesity paradox, studies that focus on pharmacological treatment as a primary objective are needed.

## 5. Conclusions

Low BMI is associated with poorer survival in patients with COPD. The mechanism underlying this observation remains unclear, although we do know that it is not because these patients experience more exacerbations and episodes of pneumonia or consume more health care resources. Nevertheless, it is important to include BMI as part of an integral evaluation of patients with COPD since this can provide useful information on prognosis.

## Figures and Tables

**Figure 1 jcm-09-00710-f001:**
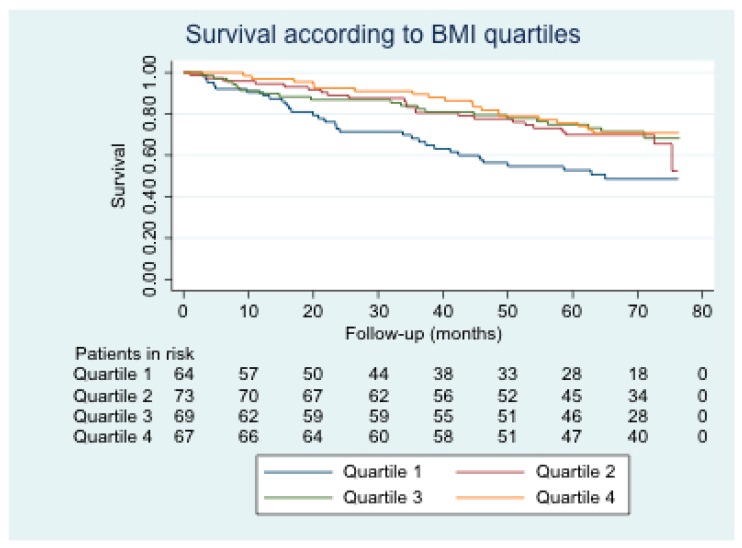
Overall survival according to the presence or absence of pneumonia.

**Table 1 jcm-09-00710-t001:** General characteristics of the patients included in the study.

Variable	Value
Patients, *n*	273
Follow-up, months (IQR)	68.15 (40.69–72.12)
Men, *n* (%)	243 (89%)
Age, years (SD)	67.99 (10.62)
Weight, kg (SD)	75.03 (16.89)
Height, m (SD)	1.63 (0.08)
BMI, kg/m^2^ (SD)	28.05 (5.49)
FEV_1_, % (SD)	48.64 (12.59)
FVC, % (SD)	73.18 (15.00)
Active smoking, *n* (%)	92 (34%)
Phenotypes, *n* (%)	
Positive bronchodilator response	71 (26.0)
Exacerbator with emphysema	27 (9.9)
Exacerbator with chronic bronchitis	40 (14.7)
Nonexacerbator	135 (49.5)
Comorbidity indices, median (IQR)	
Charlson	2 (1–4)
COTE	1 (0–2)
Pharmacological treatment, *n* (%)	
LAMA	254 (93.0)
LABA	242 (88.6)
ICS	212 (77.7)
Respiratory therapies, *n* (%)	
LTOT	91 (33.3)
CPAP	31 (11.4)
BiPAP	14 (5.1)
Events during follow-up, *n* (%)	
Exacerbation	178 (65.2)
Pneumonia	77 (28.2)
Visit to the emergency department	120 (44.0)
Hospital admission	150 (54.9)
Admission to ICU	28 (10.3)
Number of events, median (IQR)	
Exacerbation	1 (0–1)
Pneumonia	0 (0–1)
Visit to the emergency department	0 (0–1)
Hospital admission	0 (0–1)
Admission to ICU	0 (0–0)
Death, *n* (%)	93 (34.1)

IQR: interquartile range; SD: standard deviation; BMI: body mass index; FEV_1_: forced expiratory volume in the first second; FVC: forced vital capacity; COTE: COPD-specific Comorbidity Test; LAMA: long-acting muscarinic antagonist; LABA: long-acting beta2-adrenergic agonist; ICS: inhaled corticosteroid; LTOT: long-term home oxygen therapy; CPAP: continuous positive airway pressure; BiPAP: bilevel positive airway pressure; ICU: intensive care unit.

**Table 2 jcm-09-00710-t002:** Sample size, mean, and standard deviation of the BMI quartiles.

Quartile	Quartile 1	Quartile 2	Quartile 3	Quartile 4
Sample size, *n*	64	73	69	67
Mean BMI, kg/m^2^	21.27	25.95	29.50	35.33
Standard deviation	1.94	1.06	1.02	3.58

**Table 3 jcm-09-00710-t003:** Anthropometric data, lung function, phenotype, treatment, and comorbidity indexes according to BMI quartiles.

Variable	Quartile 1	Quartile 2	Quartile 3	Quartile 4	*p*-Value
Men, *n* (%)	54 (84.4)	62 (84.9)	66 (95.7)	61 (91.0)	0.110
Age, years (SD)	68.02 (11.10)	67.92 (10.66)	70.14 (10.09)	65.82 (10.42)	0.130
Pulmonary function, mean (SD)					
FEV_1_ in mL	1094 (392)	1219 (381)	1240 (456)	1327 (410)	0.015*
FEV_1_ in %	45.5 (13.5)	48.6 (12.8)	49.2 (13.1)	51.1 (10.4)	0.083
FVC in mL	2342 (667)	2428 (636)	2292 (711)	2384 (688)	0.669
FVC in %	75.5 (16.9)	74.9 (13.5)	70.4 (16.5)	72.0 (12.4)	0.146
Phenotypes, *n* (%)					
Positive bronchodilator response	14 (21.9)	22 (30.1)	18 (26.1)	17 (25.4)	0.041*
Exacerbator with emphysema	10 (15.6)	5 (6.8)	5 (7.2)	7 (10.4)	
Exacerbator with chronic bronchitis	8 (12.5)	18 (24.7)	11 (15.9)	3 (4.5)	
Nonexacerbator	32 (50.0)	28 (38.4)	35 (50.7)	40 (59.7)	
Active smoking, *n* (%)	32 (50.0)	24 (32.9)	15 (21.7)	21 (31.3)	0.001**
OSA, *n* (%)	0 (0.0)	1 (1.4)	11 (15.9)	27 (40.3)	0.000***
Pharmacological treatment, *n* (%)					
LAMA	60 (93.8)	67 (91.8)	66 (95.7)	61 (91.0)	0.713
LABA	58 (90.6)	66 (90.4)	60 (87.0)	58 (86.6)	0.811
ICS	48 (75.0)	57 (78.1)	54 (78.3)	53 (79.1)	0.948
Roflumilast	0 (0.0)	4 (5.5)	1 (1.4)	2 (3.0)	0.207
Theophylline	3 (4.7)	4 (5.5)	5 (7.2)	3 (4.5)	0.891
Respiratory therapies, *n* (%)					
LTOT	15 (23.4)	25 (34.2)	27 (39.1)	24 (35.8)	0.253
CPAP	0 (0.0)	1 (1.4)	10 (14.5)	20 (29.9)	0.000***
BiPAP	1 (1.6)	2 (2.7)	3 (4.3)	8 (11.9)	0.029*
Comorbidity indexes, median (IQR)					
Charlson	2 (1–3)	2 (1–3)	2 (1–4)	2 (1–4)	0.203
COTE	1 (0–2)	0 (0–2)	1 (0–3)	1 (0–2)	0.946

BMI: body mass index; SD: standard deviation; FEV_1_: forced expiratory volume in the first second; FVC: forced vital capacity; OSA: obstructive sleep apnea; LAMA: long-acting muscarinic antagonist; LABA: long-acting beta2-adrenergic agonist; ICS: inhaled corticosteroid; LTOT: long-term home oxygen therapy; CPAP: continuous positive airway pressure; BiPAP: bilevel positive airway pressure; IQR: interquartile range; COTE: COPD-specific Comorbidity Test; * *p* < 0.05; ** *p* < 0.01; *** *p* < 0.001.

**Table 4 jcm-09-00710-t004:** Exacerbation, pneumonia, and use of health resources during follow-up according to BMI quartiles.

Variable	Quartile 1	Quartile 2	Quartile 3	Quartile 4	*p*-Value
Events during follow-up, *n* (%)					
Exacerbation	37 (57.8)	49 (67.1)	50 (72.5)	42 (62.7)	0.327
Pneumonia	14 (21.9)	22 (30.1)	22 (31.9)	19 (28.4)	0.601
Visit to the emergency department	28 (43.8)	36 (49.3)	31 (44.9)	25 (37.3)	0.556
Hospital admission	31 (48.4)	40 (54.8)	43 (62.3)	36 (53.7)	0.449
Admission to ICU	5 (7.8)	10 (13.7)	6 (8.7)	7 (10.4)	0.673
Number of events, median (IQR)					
Exacerbation	1 (0–4)	1 (0–3.5)	1 (0–4.5)	1 (0–4)	0.816
Pneumonia	0 (0–0)	0 (0–1)	0 (0–1)	0 (0–1)	0.578
Visit to the emergency department	0 (0–1)	0 (0–2)	0 (0–1)	0 (0–1)	0.765
Hospital admission	0 (0–4)	1 (0–4)	1 (0–3.5)	1 (0–3)	0.910
Admission to ICU	0 (0–0)	0 (0–0)	0 (0–0)	0 (0–0)	0.651

ICU: intensive care unit; IQR: interquartile range.

**Table 5 jcm-09-00710-t005:** Survival of each BMI quartile, log-rank, and *p*-value of quartile 1 compared to the other quartiles.

	Quartile 1	Quartile 2	Quartile 3	Quartile 4
Survival, months (SD)	51.7 (3.5)	62.7 (2.6)	63.1 (2.9)	65.7 (2.3)
Log-rank	-	5.546	6.223	8.371
*p*-value	-	0.019*	0.013*	0.004**

* *p* < 0.05; ** *p* < 0.01.

**Table 6 jcm-09-00710-t006:** Risk factors independently associated with mortality.

Variable	Hazard Ratio	95% CI	*p*-Value
Age	1.06	1.03–1.09	0.000***
Men	2.01	0.77–5.24	0.153
FEV_1_ (in 100 mL)	0.93	0.86–0.99	0.043*
Phenotypes			
Positive bronchodilator response	0.60	0.31–1.17	0.132
Exacerbator with emphysema	1.09	0.57–2.09	0.797
Exacerbator with chronic bronchitis	1.76	1.01–3.06	0.046*
Active smoking	1.21	0.72–2.03	0.462
Charlson	1.32	1.18–1.49	0.000***
BMI quartiles			
Quartiles 1	1.99	1.08–3.69	0.028*
Quartiles 2	1.03	0.55–1.93	0.925
Quartiles 3	0.80	0.42–1.52	0.494

* *p* < 0.05; *** *p* < 0.001.

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
