# Peer review of "Differences in the Outcome of Patients with COPD according to Body Mass Index"

_jcm, 2020, doi:10.3390/jcm9030710_

Round 1

Reviewer 1 Report

The manuscript described the reason why obesity behaves as a protective factor in patients with COPD. A large number of COPD patients are suffering from chronic and serious complaints. The authors have revealed the correlation between COPD and body mass index (BMI). Thus, these findings will be useful for the remedy of COPD. Therefore, the manuscript is not too excellent to be published. In other words, the manuscript is so excellent that it should be published. Comments (1) How do the values of BMI categorized into Quartile 1, Quartile 2, Quartile 3, and Quartile 4 show the correlation to COPD? Simply stated, does that mean survival was lower for patients in BMI quartile 1 than for those in other quartiles (Figure 1)? (2) Why is low BMI associated with poorer survival in patients with COPD? (3) Are some types of hormones that relieve COPD released in high BMI persons? That is all.

Author Response

Response to Reviewer 1 Comments

Point 1: How do the values of BMI categorized into Quartile 1, Quartile 2, Quartile 3, and Quartile 4 show the correlation to COPD? Simply stated, does that mean survival was lower for patients in BMI quartile 1 than for those in other quartiles (Figure 1)?

Response 1: The mean BMI value was used to describe this variable in each comparison group. This value in each quartile is shown in Table 2: 21.27, 25.95, 29.50 and 35.33 kg/m2, respectively. Comparison groups were established based on the BMI quartiles to achieve groups with similar sample size. Patients in the BMI quartile 1 had lower survival compared to patients in quartiles 2-4, which is the main finding in our study, described in the discussion and conclusions sections.

Point 2: Why is low BMI associated with poorer survival in patients with COPD?

Response 2: There are several theories about the cause of this paradox, which are shown in the discussion section (paragraphs 3-5 of this section), but today there is no theory accepted by the majority. One possibility is that lung function is poorly estimated in patients with low BMI. Another possibility is that abdominal fat limits pulmonary hyperinflation in patients with COPD. The third possibility is that there is actually a lower muscle mass in patients with low BMI. In our study, we were unable to identify the exact mechanism by which this paradox occurs, but we know that it is not because patients with low BMI have more adverse events, and that low BMI is per se a risk factor for mortality.

Point 3: Are some types of hormones that relieve COPD released in high BMI persons? That is all.

Response 3: This proposal is very interesting. Unfortunately, this study did not collect data related to hormones, due to the design of this study (the patients included in the study were those who attended a spirometry test, not in a consultation where there are possibilities to request a blood test). It is logical to think that there could be some relationship with hormones, and in general, with proteins that have systemic effect, including pro-inflammatory factors. In fact, in another study that we have recently launched, we are collecting data on hormones and inflammation proteins to continue clearing up this paradox.

Reviewer 2 Report

I am honoured to review the manuscript entitled Differences in the outcome of patients with COPD according to body mass index which concerns a very actual and a very interesting topic.

The Abstract is correct.
The Introduction is short but essential. The aim of the paper is clear.

Methodology of the paper is described in a clear way. But I have got the few specific comments:

  1. But I don’t understand why in Design section you write: “We excluded patients with ….. those with other nonobstructive respiratory diseases…” and in the follow sections you showed patients with OSA – maybe it is worth to described it better? I am not convinced that it can be simplified that OSA is an obstructive respiratory disease.
  2. In line 101-103 I’m not sure what authors meant? Is the matter that you did not include ACO patients to the study?

Results are clearly explained, but:

  1. In Results section we have got information about treatment, it may be worth specifying at what moment patients received this treatment
  2. It may be worth writing how many patients received triple therapy or only LABA/LAMA therapy
  3. in relation to pneumonia, in what percentage were patients receiving ICS
  4. The same information about the characteristics of the group is contained in Table 1 as well as in the main text, it is better to choose one option to describe the group. I propose to delete extensive Table 1 the more that the information is well described in the text

Discussion is concise.

The conclusions are adequate.

Author Response

Response to Reviewer 2 Comments

Point 1: But I don’t understand why in Design section you write: “We excluded patients with ….. those with other nonobstructive respiratory diseases…” and in the follow sections you showed patients with OSA – maybe it is worth to described it better? I am not convinced that it can be simplified that OSA is an obstructive respiratory disease.

Response 1: In our study, we excluded patients with non-obstructive pathologies (in other words, restrictive pathologies) so that these diseases, of the pulmonary parenchyma, that suppose an alteration of the pulmonary function, with the aim that the pulmonary function only reflects the affectation by COPD. In the case of OSA, it is an obstructive disease, but not of the lung parenchyma, but of the upper airway, which does not imply an alteration in pulmonary function.

Taking into account the reviewer's comment, we point out in the Design section that excluded non-obstructive pathologies refer to pathologies of the pulmonary parenchyma.

Point 2: In line 101-103 I’m not sure what authors meant? Is the matter that you did not include ACO patients to the study?

Response 2: We have actually included patients with ACO, but with a slightly different definition. The Spanish guide defines ACO by the presence of an increase in FEV1 of more than 400 mL in absolute value and 15% in relative value after the administration of bronchodilators. We believe that these patients are rather asthmatic than COPD. In our study, "ACO" patients have an FEV1 increase of more than 200 mL in absolute value and 12% in relative value after bronchodilator administration. In this way, these are COPD patients but with bronchial hyperreactivity. Therefore, instead of defining them as "ACO", we have used the term "positive bronchodilator response".

Point 3: In Results section we have got information about treatment, it may be worth specifying at what moment patients received this treatment.

Response 3: Taking into account the reviewer's comment, we specify the variables collected at the time of inclusion in the Variables collected section (not in the Results section because we will modify this paragraph by subsequent comments):

“Variables were collected at 2 time points. The first was at recruitment, and the variables recorded were patient characteristics (including comorbidity indices, pharmacological treatment and respiratory therapies).”

Point 4: It may be worth writing how many patients received triple therapy or only LABA/LAMA therapy.

Response 4: Unfortunately, the variables on combined therapy were not collected, but only each therapy separately. Taking into account the reviewer's comment, we add a paragraph in the Discussion section mentioning this limitation.

“This study has a limitation. Regarding the pharmacological treatment, no data on the combined therapy were collected, but each therapy separately. This has only allowed us to make an analysis of each component of the treatment, but not the combination of them.”

Point 5: In relation to pneumonia, in what percentage were patients receiving ICS.

Response 5: Pneumonia in patients with COPD is an interesting and broad topic. We have addressed this topic in our previous publication. We have not mentioned detailed data on pneumonia in this work because the statistical analysis is different (it is a time dependent variable). However, we attach the reference of our previous publication below.

Ji, Z.; Hernández Vázquez, J.; Bellón Cano, J.M.; Gallo González, V.; Recio Moreno, B.; Cerezo Lajas, A.; Puente Maestu, L.; de Miguel Díez, J. Influence of Pneumonia on the Survival of Patients with COPD. J. Clin. Med. 20209, 230.

Point 6: The same information about the characteristics of the group is contained in Table 1 as well as in the main text, it is better to choose one option to describe the group. I propose to delete extensive Table 1 the more that the information is well described in the text

Response 6: Taking into account the reviewer's comment, we delete some of the information given in the text in the Results section, maintaining table 1. We consider that the table provides a faster and more comfortable reading to understand the data. In this way, the text only offers basic data, and the table offers complete data.